# Economic evaluation of prenatal screening for fetal aneuploidies in Thailand

**Preechaya Wongkrajang**[1,2], **Jiraphun Jittikoon**[3], **Wanvisa Udomsinprasert**[3],
**Pattarawalai Talungchit**[4,5], **Sermsiri Sangroongruangsri**[6], **Saowalak Turongkaravee**[6],
**Usa Chaikledkaew**[5,6]*

1 Social, Economic and Administrative Pharmacy (SEAP) Graduate Program, Faculty of Pharmacy, Mahidol University, Bangkok, Thailand, 2 Department of Clinical Pathology, Faculty of Medicine Siriraj Hospital, Mahidol University, Bangkok, Thailand, 3 Department of Biochemistry, Faculty of Pharmacy, Mahidol University, Bangkok, Thailand, 4 Department of Obstetrics and Gynecology, Faculty of Medicine Siriraj Hospital, Mahidol University, Bangkok, Thailand, 5 Mahidol University Health Technology Assessment (MUHTA) Graduate Program, Mahidol University, Bangkok, Thailand, 6 Social and Administrative Pharmacy Division, Department of Pharmacy, Faculty of Pharmacy, Mahidol University, Bangkok, Thailand

* usa.chi@mahidol.ac.th

## Abstract

Historically, there has been a lack of cost-effectiveness data regarding the inclusion of universal non-invasive prenatal testing (NIPT) for trisomy 21, 18, and 13 in the benefit package of the Universal Health Coverage (UHC) in Thailand. Therefore, this study aimed to perform the cost-benefit analysis of prenatal screening tests and calculate the budget impact that would result from the implementation of a universal NIPT program. A decision-tree model was employed to evaluate cost and benefit of different prenatal chromosomal abnormalities screenings: 1) first-trimester screening (FTS), 2) NIPT, and 3) definitive diagnostic (amniocentesis). The comparison was made between these screenings and no screening in three groups of pregnant women: all ages, < 35 years, and ≥ 35 years. The analysis was conducted from societal and governmental perspectives. The costs comprised direct medical, direct non-medical, and indirect costs, while the benefit was cost-avoidance associated with caring for children with trisomy and the loss of productivity for caregivers. Parameter uncertainties were evaluated through one-way and probabilistic sensitivity analyses. From a governmental perspective, all three methods were found to be cost-beneficial. Among them, FTS was identified as the most cost-beneficial, especially for pregnant women aged ≥ 35 years. From a societal perspective, the definitive diagnostic test was not cost-effective, but the other two screening tests were. The most sensitive parameters for FTS and NIPT strategies were the productivity loss of caregivers and the incidence of trisomy 21. Our study suggested that NIPT was the most cost-effective strategy in Thailand, if the cost was reduced to 47 USD. This evidence-based information can serve as a crucial resource for policymakers when making informed decisions regarding the allocation of resources for prenatal care in Thailand and similar context.

**Data Availability Statement:** All relevant data are within the paper.

**Funding:** This study receives funding support from the Health Systems Research Institute (HSRI). The funders had no role in study design, data collection and analysis, decision to publish, or preparation of the manuscript.

**Competing interests:** The authors have declared that no competing interests exist.

# Introduction

Chromosomal abnormalities affect approximately 0.5 and 1.0% of live births [1]. Trisomy 21 (T21), often known as Down syndrome, is the most common of these abnormalities, occurring one in every 700 live births [1]. The average life expectancy of individuals with Down syndrome has increased by 50 years [2]. This syndrome imposes a significant economic burden due to its long-life expectancy and diverse clinical complications including developmental delays, intellectual disabilities, cardiovascular diseases, and gastrointestinal defects [3, 4]. Trisomy 18 (T18) or Edwards syndrome, is the second most prevalent chromosomal abnormality, with a prevalence of one in every 3,000 to 8,000 live births [1, 5]. Patau syndrome, also known as trisomy 13 (T13), is a relatively prevalent trisomy condition, occurring in approximately one out of every 12,000 live births [1]. Over 95% of these fetuses experience spontaneous miscarriage. The survival rate for infants diagnosed with T18 is approximately 1%, while that for those diagnosed with T13 is approximately 10% within their first year of life [1, 5]. These trisomy patients require a multidisciplinary team of healthcare providers to manage their health conditions, resulting in a high economic burden, especially in the case of children [6–13].

Prenatal screening tests, including serum screening, nuchal translucency (NT) from ultrasound, and genetic screening, are currently essential for assessing the potential risk of an unhealthy fetus [14]. The first-trimester screening test (FTS) for detecting T13, T18, and T21 conducted between 10 and 13 weeks of gestational age [14] involves measuring pregnancy-associated plasma protein A and free β-human chorionic gonadotropin levels, as well as performing an ultrasound to measure NT. An additional prenatal screening test available is the quadruple test, which can be typically performed during the second trimester between 15 and 22 weeks of gestational age. This test involves measuring four serum markers: α-fetoprotein, free β-human chorionic gonadotropin, unconjugated estriol, and inhibin A and can only screen for T18 and T21 [14]. The trisomy risk assessment for these screening tests incorporates data from serum tests, NT results, and additional factors including age, weight, race, and gestation age of pregnant women [14]. According to previous studies, the detection rate for FTS have been observed to vary between 71.9% and 84% for T13, 71.9% and 97% for T18, and 71.4% and 91.7% for T21, with the false positive ranging from 0.5% to 7% [15–19]. For the quadruple test, the detection rate is around 80% for T18, ranging from 67% to 76.2% for T21, and the false positive varies between 5% to 14% [16, 20–23]. In Thailand, the costs of FTS and the quadruple test are estimated to be approximately 30 Unites States dollars (USD), with a typical turnaround time of approximately one week [24]. It is noteworthy that several factors could contribute to a high false positive rate in serum screening test. However, in recent years, a non-invasive prenatal testing (NIPT) has been developed to detect fetal cell-free DNA (cfDNA) in maternal plasma. This method has proven to be effective in accurately identifying chromosomal abnormalities in the in the fetus [14].

NIPT is commonly used in clinical practice due to its accuracy with a detection rate of more than 99% for T21, 90% for T18, and 60% for T13, resulting in a false-positive rate of less than 1% [23]. However, NIPT incurs a cost of approximately 300 USD and entails a turnaround time of approximately two weeks [24]. In cases where screening test results indicate a high risk, pregnant women are provided with the option of undergoing definitive diagnostic testing to confirm the presence of a chromosomal anomaly in the fetus. This can be done through procedures such as chorionic villus sampling (CVS) or amniocentesis. It is important to note that this recommendation applies to all pregnant women, regardless of their age or the perceived risk of chromosomal abnormality [14]. Currently, NIPT is considered a costly technology, and as a result, it has not been incorporated into the benefit package provided by the Universal Health Coverage (UHC), which covers approximately 80% of the Thai population.

Although two previously published studies have been available [25, 26], it is worth noting that these studies specifically focused on evaluating the cost-benefit analysis of NIPT for Down syndrome (T21). However, they did not encompass the evaluation of Edwards (T13) and Patau syndromes (T18), which are the second and third most prevalent trisomy, respectively.

Since 2016, the Subcommittee on Health Promotion and Disease Prevention under the National Health Security Office (NHSO) has included Down syndrome screening in the UHC's benefit package for pregnant women aged 35 years and over [27]. Later since 2022, the Subcommittee has implemented an expansion of the quadruple test for pregnant women across all age groups, commonly referred to quadruple for all [27]. Consequently, the implementation of Down Syndrome Prevention and Control Pilot Program in 2016 across five provinces yielded a high acceptance rate of quadruple screening, but resulted in a false positive rate of 4–10%, potentially leading to a considerable number of unnecessary amniocentesis procedures [28]. As of the time, there is a lack of available cost-effectiveness data regarding the optimal prenatal screening test for T13, T18, and T21, as well as the recommended age group of pregnant women for screening. Therefore, the NHSO has formally requested the aforementioned information to facilitate informed decision-making regarding the potential implementation of a universal prenatal screening test policy, which would encompass all pregnant women. Accordingly, the objective of this study was to conduct a cost-benefit analysis of different prenatal chromosomal abnormality screening tests for detecting T13, T18, and T21 in all pregnant women classified into three age groups: all ages, older than 35 years, and less than 35 years.

## Materials and methods

### Target population

The cost-benefit analysis was performed using a decision-analytic model to calculate the costs and benefits of different screening strategies based on governmental and societal perspectives using a lifetime period. In addition, target populations consisted of pregnant women who were classified into three groups: all ages, those older than 35 years, and those younger than 35 years. These three groups with a cut-off age at 35 years were selected for evaluating each screening strategy based on the recommendation of the American College of Obstetricians and Gynecologists Clinical Practice Guidelines [29]. This study was approved by Siriraj Institutional Review Board (SIRB) (MU-MOU COA 657/2021) and the ethics committee waived the requirement for informed consent.

### Model structure

A decision-analytic model was developed to conduct a cost-benefit analysis of three screening strategies: 1) FTS, 2) NIPT, and 3) definitive diagnosis, in comparison to no screening. Regarding FTS option (Fig 1), pregnant women are provided with the opportunity to make an informed decision regarding their acceptance or declination of the FTS test. If the test is accepted, it is possible for the results to detect the presence of T13, T18, and T21 in the fetuses, either through true positive or false negative results. If the individuals received true positive test results and accepted definitive diagnosis, they might have procedure related abortion. For those with true positive results, they might choose to undergo pregnancy termination, deliver live births with chromosomal abnormalities such as T13, T18, and T21, or experience spontaneous abortion. For those with false negative results or those denying definitive diagnosis, they might deliver live births with T13, T18, and T21 or have spontaneous abortion. On the other hand, for those having non-trisomy detected by false positive result, they would choose to either accept or reject the definitive diagnosis. For those with true negative results, they would

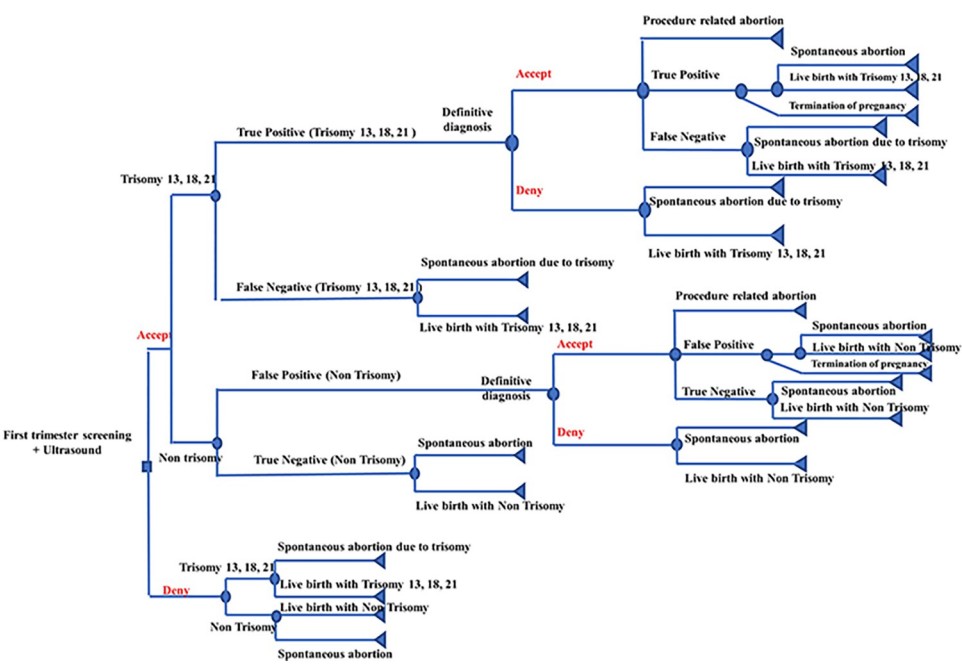

**Fig 1. Decision tree model for first-trimester screening testing.**

either give birth to infants without trisomy or experience spontaneous abortion. However, in the event that they decline the FTS test, there is a possibility of delivering live births without trisomy or experiencing spontaneous abortion.

In the context of a universal NIPT alternative (Fig 2), pregnant women would have the option to either accept or deny the utilization of NIPT. If the NIPT results were positive, it would be possible that the fetuses might have been identified with T13, T18, and T21, or non-trisomy conditions, as determined by either successful or unsuccessful NIPT outcomes. For those with NIPT failure, they would be provided with a conclusive diagnosis. In cases where NIPT yields successful results, it is possible to detect fetuses with T13, T18, and T21 through either false positive or true negative finding. Conversely, in instances where NIPT is unsuccessful, it is possible to detect non-trisomy fetuses through either false positive or true negative finding. The decision tree model pathways for definitive diagnosis, true positive, false negative, false positive, and true negative in NIPT were found to be comparable to those observed in FTS. Fig 3A and 3B demonstrates definitive diagnosis or amniocentesis for T13, T18, and T21 as well as non-trisomy, respectively. The data analysis was performed using Microsoft Excel 2019 (Microsoft, WA, USA).

## Model parameters

The parameters used in the model are presented in Table 1. Parameters were obtained from Siriraj hospital's database, published articles, and nationwide and regional public data in 2019.

## Probabilities

The probabilities related to trisomy incidence, miscarriage, rate of pregnancy termination, uptake rate of screening test, uptake rate of diagnostic test following a high-risk result of screening test, failure rate of NIPT, and procedure-related miscarriage were retrieved from Siriraj hospital's databases and published articles [25, 30, 35]. The sensitivity and specificity of

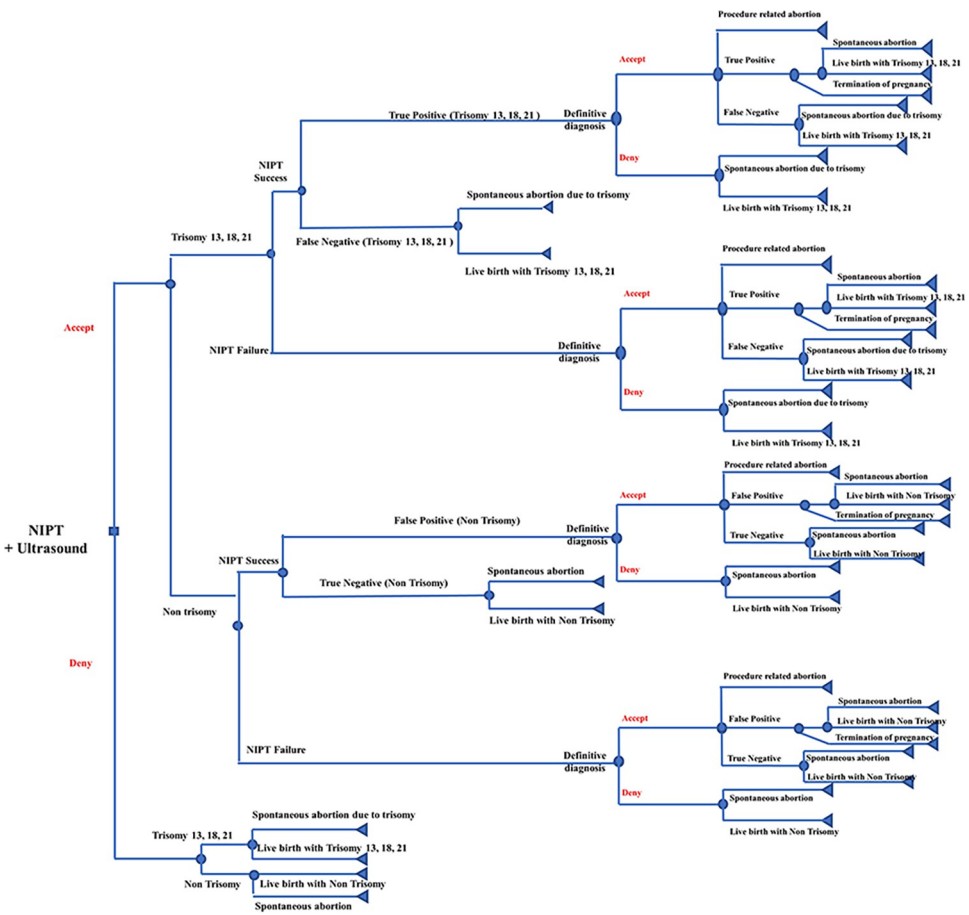

**Fig 2. Decision tree model for non-invasive prenatal testing.**

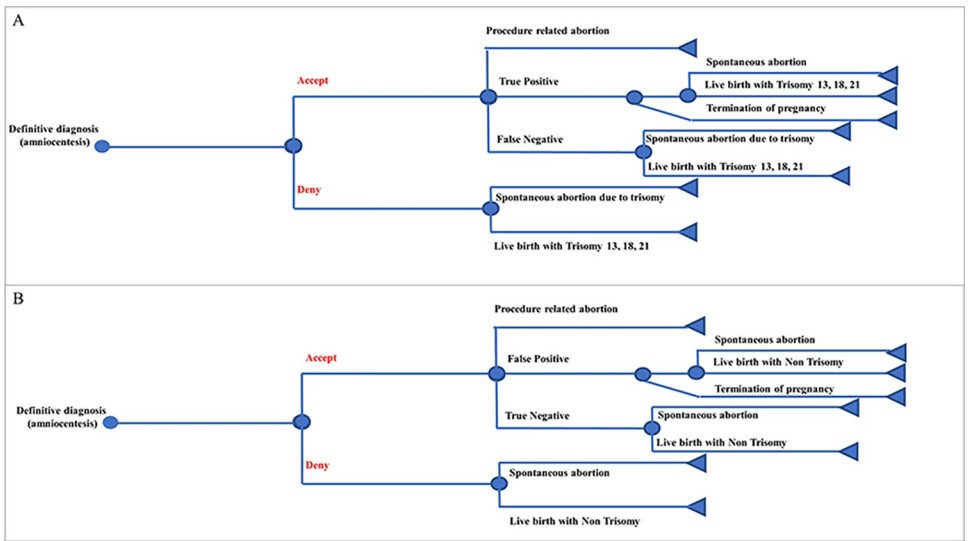

**Fig 3. Decision tree model for definitive diagnosis** A) Trisomy 13, 18, 21 B) Non-Trisomy.

**Table 1. Parameters used in this study.**

| Parameter | Distribution* | Mean | Standard Error | Source |
|---|:---:|:---:|:---:|:---:|
| **Probability** | | | | |
| Trisomy 13 | | | | |
| Incidence in women aged <35 years | Beta | 0.0004 | 0.000038 | Siriraj hospital |
| Incidence in women aged > 35 years | Beta | 0.0025 | 0.000254 | Siriraj hospital |
| Incidence in all women | Beta | 0.0011 | 0.000105 | Siriraj hospital |
| Trisomy 18 | | | | |
| Incidence in women aged <35 years | Beta | 0.0000 | 0.000000 | Siriraj hospital |
| Incidence in women aged > 35 years | Beta | 0.0025 | 0.000254 | Siriraj hospital |
| Incidence in all women | Beta | 0.0008 | 0.000079 | Siriraj hospital |
| Trisomy 21 | | | | |
| Incidence in women aged <35 years | Beta | 0.0019 | 0.000191 | Siriraj hospital |
| Incidence in women aged > 35 years | Beta | 0.0102 | 0.001015 | Siriraj hospital |
| Incidence in all women | Beta | 0.0045 | 0.000448 | Siriraj hospital |
| Incidence of miscarriage | | | | |
| All women | Beta | 0.0635 | 0.006350 | Siriraj hospital |
| Trisomy 13 | Beta | 0.1200 | 0.012000 | [30] |
| Trisomy 18 | Beta | 0.2000 | 0.020000 | [30] |
| Trisomy 21 | Beta | 0.2713 | 0.027130 | [30] |
| Rate of pregnancy termination Trisomy 13 | LOG Normal | 0.5000 | 0.050000 | Siriraj hospital |
| Rate of pregnancy termination Trisomy 18 | LOG Normal | 0.9900 | 0.099000 | Siriraj hospital |
| Rate of pregnancy termination Trisomy 21 | LOG Normal | 0.9412 | 0.094120 | Siriraj hospital |
| Rate of procedure-related miscarriage | Beta | 0.002 | 0.00020 | Siriraj hospital |
| **The uptake rate of test** | | | | |
| First-trimester screening test | Beta | 0.9557 | 0.095570 | [26] |
| Non-invasive prenatal test (NIPT) | Beta | 0.9245 | 0.092450 | [26] |
| Diagnostic test | Beta | 0.9500 | 0.095000 | assumption for free of charge |
| The uptake rate of diagnostic tests after high-risk results in the first trimester | Beta | 0.9000 | 0.090000 | Siriraj hospital |
| The uptake rate of diagnostic tests after high-risk results in NIPT | Beta | 0.9000 | 0.090000 | Siriraj hospital |
| The failure rate of NIPT | Beta | 0.0222 | 0.002217 | Siriraj hospital |
| The uptake rate of diagnostic tests after failure results in NIPT | Beta | 0.6667 | 0.066667 | Siriraj hospital |
| **Cost (USD)** | | | | |
| **Direct medical costs** | | | | |
| Cost of an office visit with counseling per test | Gamma | 7.9 | 1.6 | Siriraj hospital |
| Cost of First-trimester screening test per test | Gamma | 24.3 | 4.9 | Siriraj hospital |
| Cost of NIPT per test | Gamma | 237.0 | 47.4 | Siriraj hospital |
| Cost of ultrasound per test | Gamma | 23.6 | 4.7 | Siriraj hospital |
| Cost of Diagnostic test: Invasive procedure per test | Gamma | 141.5 | 28.3 | Siriraj hospital |
| Cost of medical services procedure-related loss per visit | Gamma | 77.2 | 15.4 | Siriraj hospital |
| Cost of elective termination per visit | Gamma | 154.4 | 30.9 | Siriraj hospital |
| Cost of normal labor per visit | Gamma | 218.8 | 43.8 | Siriraj hospital |
| Cost of Cesarean section per visit | Gamma | 308.8 | 61.8 | Siriraj hospital |
| Cost of termination of pregnancy/miscarriage per visit | Gamma | 257.4 | 51.5 | Siriraj hospital |
| **Direct non-medical costs** | | | | |
| Cost of travel per visit | Gamma | 4.0 | 0.8 | [31] |
| Cost of food per visit | Gamma | 1.5 | 0.3 | [31] |
| The opportunity cost of pregnant women per visit | Gamma | 2.2 | 0.4 | [31] |
| The opportunity cost of a caregiver per visit | Gamma | 2.7 | 0.5 | [31] |

*(Continued)*

**Table 1.** (Continued)

| Parameter | Distribution* | Mean | Standard Error | Source |
|---|---|---|---|---|
| **Indirect cost** | | | | |
| Loss from miscarriage due to definitive diagnosis or termination in normal case | Gamma | 449,357.7 | 89,871.5 | Calculated from GDP [32] |
| **Benefit (USD)** | | | | |
| **Direct benefit** | | | | |
| Cost avoidance of trisomy 13 | Gamma | 34,517.6 | 6,903.5 | [33] |
| Cost avoidance of trisomy 18 | Gamma | 34,517.6 | 6,903.5 | [33] |
| Cost avoidance of trisomy 21 | Gamma | 46,976.0 | 9,395.2 | [6] |
| **Indirect benefit** | | | | |
| Productivity loss of caregiver trisomy 13 or trisomy 18 | Gamma | 6,370.8 | 1,274.2 | Calculated from GDP [32] |
| Productivity loss of caregiver trisomy 21 | Gamma | 446,648.1 | 89,329.6 | Calculated from GDP [32] |
| **Test performance** | | | | |
| Screening tests | Distribution | Sensitivity | Specificity | Source |
| **First-trimester test** | | | | |
| Trisomy 13 | LOG Normal | 0.5000 | 1.000 | [23] |
| Trisomy 18 | LOG Normal | 0.8000 | 1.000 | [23] |
| Trisomy 21 | LOG Normal | 0.8000 | 1.000 | [23] |
| **Non-invasive prenatal testing (NIPT)** | | | | |
| Trisomy 13 | LOG Normal | 1.000 | 0.9980 | [34] |
| Trisomy 18 | LOG Normal | 1.000 | 0.9920 | [34] |
| Trisomy 21 | LOG Normal* | 1.000 | 0.9980 | [34] |
| **Diagnostic test** | | | | |
| Definitive diagnosis | LOG Normal* | 0.9900 | 0.9900 | Siriraj hospital |

*Beta, log-normal, or gamma distribution is appropriate for parameter values ranging from 0 to 1, 0 to $\infty$, and > 0 to $\infty$, respectively.

these tests were determined based on a study conducted by Badeau et al. [23] and Manotaya et al. [34].

## Costs

According to a societal perspective, the analysis considered direct medical costs, direct non-medical costs, and indirect costs, whereas only direct medical costs were incorporated based on a governmental perspective. All costs were adjusted to reflect the 2022 values by utilizing the consumer price index (CPI). Subsequently, the costs were converted from Thai baht (THB) to USD using the exchange rate of 38.08 THB per 1 USD (2022 prices). All future costs and health outcomes were adjusted to their present values using a discount rate of 3% per annum.

## Direct medical cost

Direct medical costs included screening tests comprising counseling fees, screening tests (FTS, NIPT), diagnostic tests, ultrasound examinations, medical services related to procedure-related loss, delivery procedures (elective termination, normal labor, cesarean section), and service care for termination of pregnancy or miscarriage. The cost of these parameters was extracted from Siriraj hospital's database in 2016 from the Siriraj Informatics and Data

Innovation Center (SiData+), Faculty of Medicine, Siriraj hospital and a study conducted by Wanapirak et al. [26].

### Direct non-medical cost

Direct non-medical costs, including food, travel, and opportunity costs of pregnant women and caregivers during the screening test, were obtained from the Standard Cost List for Health Technology Assessment (HTA), a recognized reference cost list in Thailand [31].

### Indirect cost

Indirect cost was a productivity loss due to miscarriage from definitive diagnosis or termination of a non-trisomy case which were calculated using a human capital approach [36]. The productivity loss or income loss was estimated by working age range multiplied by the Thai Gross Domestic Product (GDP) per capita per year (6,370.8 USD) [32]. It was assumed that the working age range was 45 years (60–15 years). In addition, based on the recommendation from the Thai HTA guidelines, since cost values are different in different time periods, future values of total expected productivity loss (FV) should be adjusted to present values (PV) using an annual discount rate of 3% [37] based on this formula: $PV = FV \times [1/(1+r)^n]$, where PV = present value, FV = future value, r = discount rate, and n = each year in the future [37]. Moreover, we also assumed that expected income was increased by 4% per year, which was obtained from an annual income growth rate during 1990–2022 in Thailand [38].

### Benefits

The total benefit was the sum of cost avoidance and productivity gain of caregivers for trisomy children.

### Direct benefit

Cost avoidance was healthcare costs which were avoided by eliminating the occurrence of trisomy children as a consequence of each screening strategy. It was assumed that the avoided healthcare costs were equal to the average total healthcare costs of patients with T13, T18, and T21 which were retrieved from previous studies by Pattanaphesaj et al. [35] and Walker et al. [33].

### Indirect benefit

An indirect benefit was the productivity gain of caregivers who did not have to take care of trisomy children as a result of prenatal screening test. Human capital approach was applied by multiplying the average expected survival of trisomy patients with the caregivers' expected income referred from the annual Thai GDP per capita per year (6,370.8 USD) [32]. We assumed that the survival of fetuses with T13 and T18 was one year [39], while that of those with T21 was 50 years [2]. For caregivers of trisomy children with T21, future values of total expected productivity gain were adjusted to their present values using the discount rate of 3% [37] and income growth rate of 4% per year [38].

### Result presentation

Results of cost-benefit analysis were presented as (1) net benefit, a difference in benefit minus a difference in cost between each screening strategy and no screening (Δ benefit-Δ cost) and (2) benefit-to-cost ratio, the division of the difference in benefit by the difference in cost (Δ benefit/Δ cost). Moreover, the results were presented in terms of the net benefit and the

benefit-to-cost ratio of each screening strategy when compared to no screening. The pregnant women were classified into three groups based on age (all ages, < 35 years, ≥35 years) from both societal and governmental perspectives.

## Uncertainty analysis

The assessment of the impact of uncertainty in each parameter on the cost-benefit analysis results was conducted through the utilization of one-way sensitivity and probabilistic sensitivity analyses. The one-way sensitivity analysis was presented as a tornado diagram, while a probabilistic sensitivity analysis was performed by the Monte Carlo simulation with 1,000 iterations. A threshold sensitivity analysis was conducted to investigate the cost-effective price of NIPT.

## Budget impact analysis

Budget impact analysis (BIA) was conducted to evaluate the financial consequences of the adoption of each screening test. The total costs for each strategy at the population level were determined by multiplying the total cost from the governmental perspective per person by the population size. This study used a target population consisting of 700,000 single pregnancies, using the average birth rate data from 1993 to 2019 in Thailand [40]. The ratio of women aged ≥ 35 years to those aged < 35 years, as obtained from Siriraj hospital, was 30:70.

# Results

## Cost-benefit analysis

Table 2 demonstrates the cost-benefit results of various screening strategies in comparison to no screening. Based on the societal perspective, the cost analysis revealed that total costs of FTS in pregnant women < 35 years, ≥ 35 years, and across all age groups were the most economical, amounting to 307 USD, 320 USD, and 362 USD, respectively. Following FTS, NIPT incurred higher costs at 451 USD, 415 USD, and 516 USD for the respective age groups. Finally, definitive diagnostic testing was found to be the most expensive option, with costs of 1,927 USD, 3,514 USD, and 2,849 USD for the three age categories, respectively. However, the individuals aged over 35 years who received a definitive diagnosis had the highest total benefits, amounting to 4,259 USD. This was followed by those who underwent NIPT with total benefits of 3,296 USD, and individuals who underwent FTS with total benefits of 3,205 USD. From a governmental perspective, similar results were observed.

Furthermore, the results showed that the highest net benefit among pregnant women aged over 35 years was observed with NIPT at 3,511 USD, followed by FTS at 2,885 USD, and definitive diagnosis at 744 USD, respectively. However, definitive diagnosis did not provide any positive net benefit for individuals under the age of 35 (-1,121 USD) and for all age groups (-953 USD). Nevertheless, the benefit-to-cost ratio of FTS, NIPT, and definitive diagnostic in those women were 10.02, 6.59 and 1.21, respectively. Moreover, based on a governmental perspective, the net benefits of FTS, NIPT, and definitive diagnostic procedures in individuals aged over 35 years were calculated to be 2,870 USD, 3,561 USD, and 3,833 USD, respectively. These calculations resulted in benefit-to-cost ratio of 10.59, 8.03 and 11.05, respectively.

## Uncertainty analysis

Results of one-way sensitivity analysis are displayed in the Tornado diagrams (Figs 4–6). The most sensitive parameters were the productivity loss of caregivers for individuals with T21 and the incidence of T21 for FTS and NIPT. However, the most sensitive parameters for definitive

**Table 2. The results of the cost-benefit analysis.**

| Societal perspective | | | | | | | | | | | | |
|---|---|---|---|---|---|---|---|---|---|---|---|---|
| Screening strategies | No screening | | | 1. First-trimester test risk cut-off of 1:250 (FTS) | | | 2. Non-invasive prenatal test (NIPT) | | | 3. Definitive diagnosis | | |
| | < 35 years | ≥ 35 years | All | < 35 years | ≥ 35 years | All | < 35 years | ≥ 35 years | All | < 35 years | ≥ 35 years | All |
| Cost (USD) | 0 | 0 | 0 | 307 | 320 | 362 | 451 | 415 | 516 | 1,927 | 3,514 | 2,849 |
| Benefit (USD) | 0 | 0 | 0 | 606 | 3,205 | 1,426 | 738 | 3,926 | 1,769 | 806 | 4,259 | 1,897 |
| Difference in cost | | | | 307 | 320 | 362 | 451 | 415 | 516 | 1,927 | 3,514 | 2,849 |
| Difference in benefit | | | | 606 | 3,205 | 1,426 | 738 | 3,926 | 1,769 | 806 | 4,259 | 1,897 |
| Net benefit | | | | 298 | 2,885 | 1,064 | 287 | 3,511 | 1,253 | (-1,121) | 744 | (-953) |
| Benefit-to-cost ratio | | | | **1.97** | **10.02** | **3.94** | **1.30** | **6.59** | **2.88** | **0.42** | **1.21** | **0.67** |
| Governmental Perspective | | | | | | | | | | | | |
| Screening strategies | No screening | | | 1. First-trimester test risk cut-off of 1:250 (FTS) | | | 2. Non-invasive prenatal test (NIPT) | | | 3. Definitive diagnosis | | |
| | < 35 years | ≥ 35 years | All | < 35 years | ≥ 35 years | All | < 35 years | ≥ 35 years | All | < 35 years | ≥ 35 years | All |
| Cost (USD) | 0 | 0 | 0 | 299 | 299 | 299 | 312 | 312 | 312 | 382 | 381 | 382 |
| Benefit (USD) | 0 | 0 | 0 | 605 | 3,170 | 1,389 | 739 | 3,873 | 1,697 | 804 | 4,214 | 1,847 |
| Difference in cost | | | | 299 | 299 | 299 | 312 | 312 | 312 | 382 | 381 | 382 |
| Difference in benefit | | | | 605 | 3,170 | 1,389 | 739 | 3,873 | 1,697 | 804 | 4,214 | 1,847 |
| Net benefit | | | | 307 | 2,870 | 1,090 | 427 | 3,561 | 1,386 | 422 | 3,833 | 1,465 |
| Benefit-to-cost ratio | | | | **2.03** | **10.59** | **4.64** | **1.53** | **8.03** | **3.52** | **2.1** | **11.05** | **4.83** |

diagnosis were the loss from miscarriage due to definitive diagnosis, uptake rate, and productivity loss of caregivers for T21. Furthermore, based on the results of probabilistic sensitivity analysis, the cost-benefit planes demonstrated that most simulations were situated in the northeast quadrant. This suggests that in order to achieve greater benefits, there would be a need for higher costs associated with FTS (Fig 7A), NIPT (Fig 7B), and definitive diagnosis (Fig 7C). According to the findings of the threshold sensitivity analysis (Table 3), it was observed that a reduction of 80% in the cost of NIPT, equivalent to 47 USD, would result in an increase in the benefit-to-cost ratio of NIPT from 6.59 to 9.27. This ratio would then be comparable to that of FTS. In addition, Table 4 shows that the implementation of NIPT with a 100% uptake rate among pregnant women in different age groups (<35 years, ≥35 years, and all ages) resulted in the highest budget allocation (118, 51, and 169 million USD) compared to the budgets allocated for FTS (24, 10, and 35 million USD) and definitive diagnosis strategy (78, 14, and 92 million USD) for the respective age groups.

## Discussion

While there have been two existing cost-benefit analysis studies conducted on NIPT for Down syndrome (T13) in Thailand, no such studies have been performed for Edwards (T18) and Patau syndrome (T21), which are the second and third most prevalent chromosomal abnormalities, respectively. Therefore, this study is the first to conduct the cost-benefit analysis of prenatal screening tests, including FTS, NIPT, and definitive diagnosis, in comparison to the absence of a screening strategy for the prevention of fetal aneuploidies for T21, T13, and T18 in Thailand. The findings of this study revealed that all three methods were cost-beneficial based on the governmental perspective. Among these methods, FTS was identified as the most cost-beneficial option, particularly for pregnant women aged over 35 years. The cost of FTS was significantly lower (27.5 USD) compared to NIPT (237 USD). However, when considering the societal perspective, it is generally not considered cost-beneficial to pursue a definitive

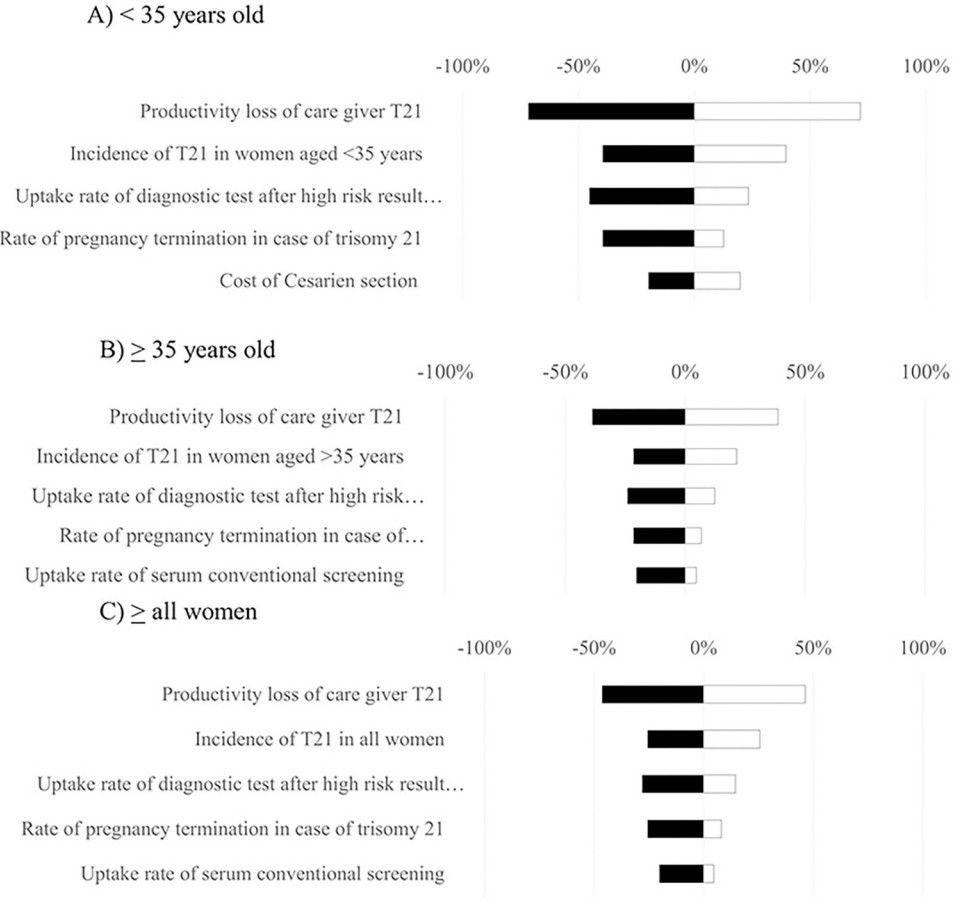

**Fig 4. Tornado diagram for first-trimester screening testing.**

diagnosis, as this often led to additional indirect costs due to the loss of a pregnancy in typical cases.

Although FTS was the most cost-beneficial and consumed the lowest budget, its practicality in the specific context of Thailand may be questionable. Given that FTS is a combined screening test for fetal NT thickness using ultrasound and serum markers, specifically free beta-hCG and PAPP-A in Thailand, there are several factors that could influence its accuracy and potentially result in a higher rate of false positive results [14]. It has been shown that the reference ranges of serum markers can be influenced by various factors, including ethnicity, as well as other variables such as gestational age, weight, smoking, and number of fetuses [41]. In addition to this, it is crucial to ensure that specimens are subjected to centrifugation within a maximum time frame of 2 hours following their collection. It is not advisable to prefer the shipment of whole blood samples due to the observed rise in specific analytes [42]. Moreover, the incorporation of NT and estimated gestational age in the risk assessment for FTS, along with the use of ultrasonography-based gestational age estimation for the quadruple test, has resulted in a detection rate of these screening tests that is notably lower than the theoretical value. This discrepancy can be attributed to the reliance on the expertise of the sonographer [43]. To ensure timely administration of FTS, it is imperative for pregnant women to promptly visit healthcare facilities within the gestational age window of 10 to 13 weeks. Furthermore, it is crucial to acknowledge that the accuracy of NT and gestational age measurements can be influenced by variables such as experience and technique. Therefore, it is essential that these

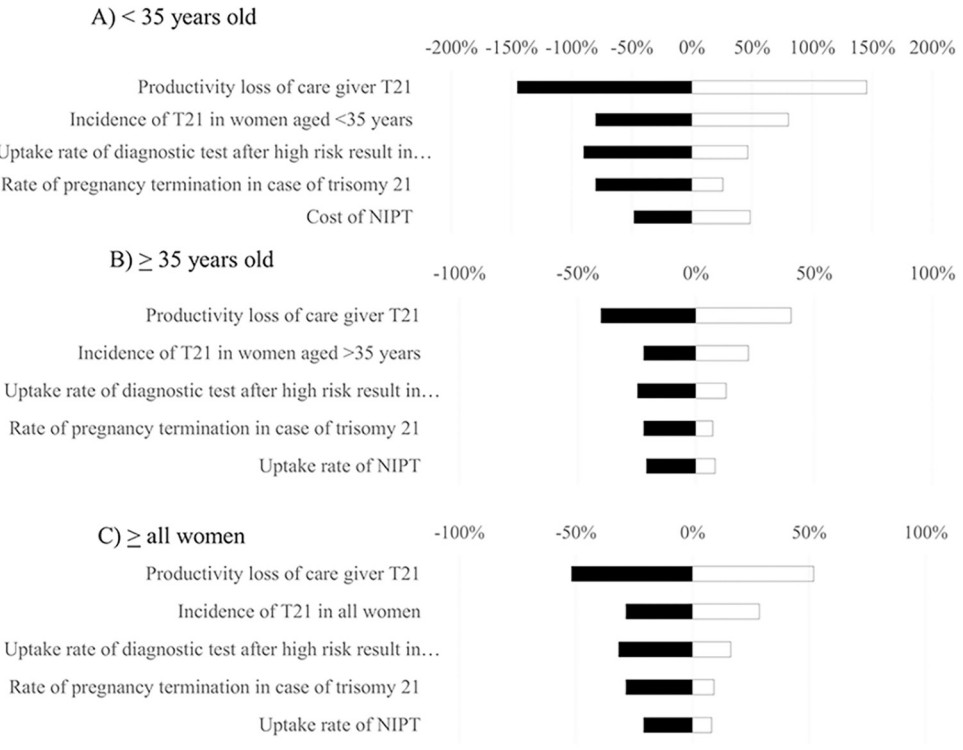

**Fig 5. Tornado diagram for non-invasive prenatal testing.**

tests are carried out by skilled sonographers. To ensure consistency and reliability, it is advisable to establish standardized measurements for NT and gestational age that are universally adopted in all healthcare facilities [43]. Moreover, risk calculation can be influenced by various factors such as the age of pregnant women, smoking habits, race, gestational age, and the requirement for specific serum transportation [42]. On the other hand, NIPT can be performed after 10 weeks of gestation until delivery, allowing pregnant individuals the flexibility to avoid early hospital visits. Consequently, NIPT exhibited not only a high level of accuracy and a low incidence of false-positive results, but also reduced vulnerability to external factors. In addition, it is worth noting that this particular test specifically focuses on evaluating maternal serum and does not necessitate the expertise of skilled sonographers [14].

Based on our research, it has been uncovered that NIPT was a cost-beneficial screening option, particularly for pregnant women aged 35 years and older, when compared to no screening based on a societal perspective. This finding was in line with a study conducted by Kostenko et al., which provided evidence supporting the cost-effectiveness of the universal NIPT for T13, T18, and T21 [44]. Similarly, the studies conducted by Walker et al. in the US, Wang et al., Shang et al., and Xiao et al. in China all provided evidence that the universal NIPT for T21 was a cost-effective approach [18, 33, 43, 45]. In contrast, previous studies conducted by Evan et al. from the US and Beulen et al. from the Netherlands both revealed that the universal NIPT was currently not regarded as a cost-effective strategy for identifying T21 in newborns. This is primarily attributed to its high cost associated with the procedure, typically ranging from 800 to 1,000 USD [46, 47]. Moreover, it is imperative to acknowledge that the aforementioned studies differ from our own study in several aspects. Specifically, they performed a cost-effectiveness analysis using a decision tree model from a payer perspective. Additionally, their assessment of effectiveness was based on the number of T21 cases [46, 47].

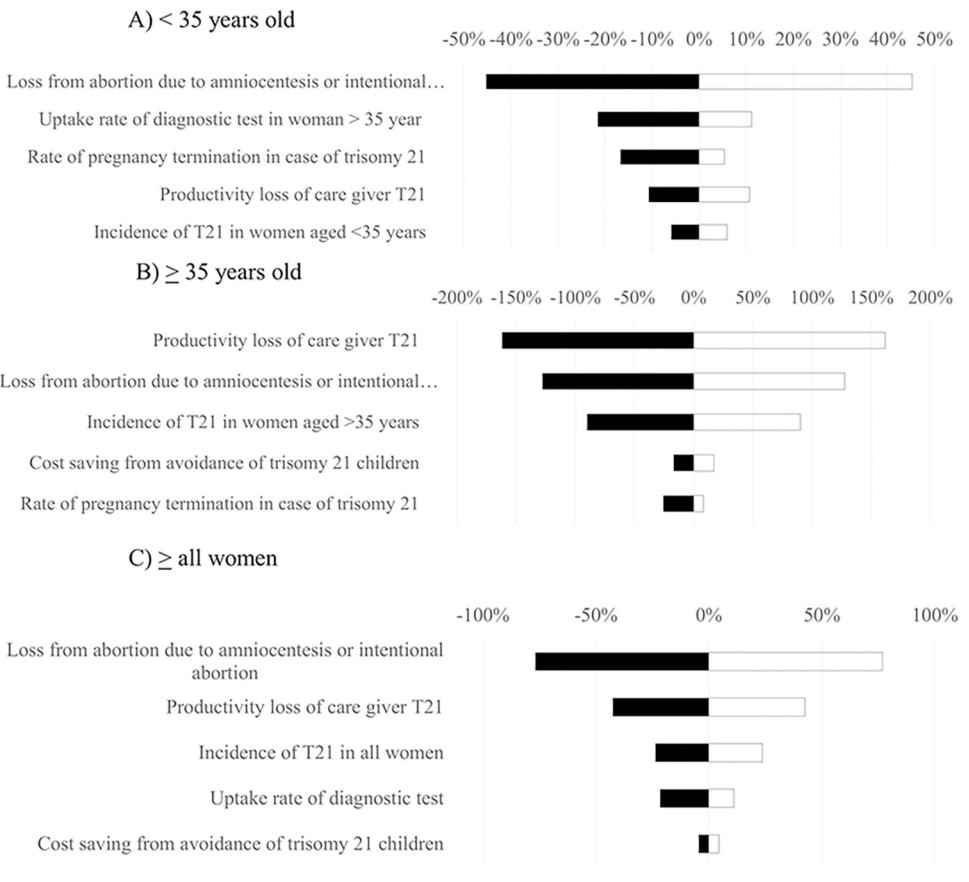

**Fig 6. Tornado diagram for definitive diagnosis.**

In contrast to our study, two previously published cost-benefit studies conducted in Thailand [25, 26] suggest that implementing a universal NIPT as the primary screening method for Down syndrome (T13) was not deemed cost-beneficial. However, these studies ascertained that utilizing NIPT as a secondary screening approach was the most cost-beneficial choice. It should be noted that the discrepancies in our results can be ascribed to the following reasons. First, we conducted an analysis on the cost-benefit of implementing a universal NIPT for Down syndrome (T13), Edwards syndrome (T18), and Patau syndrome (T21), while two aforementioned studies specifically focused on Down syndrome (T13) exclusively. Apart from this, the majority of the data, such as prevalence, incidence, uptake rate, and costs, were obtained from Siriraj Hospital, while a study by Oraluck et al. [25] primarily relied on the findings of Pattanaphesaj et al. [35], which were conducted a decade ago. Besides this, a study by Wanapirak et al. [26] primarily utilized most of the data from their previous study [22]. Consequently, our study did not coincide with the findings of the previously mentioned studies.

In addition to this, the sensitivity analysis uncovered two influential factors that should be considered. The factors taken into consideration included the productivity loss of caregivers for T21 fetuses, as well as the incidence of T21 for NIPT. The productivity loss among caregivers for T21 fetuses is contingent upon the GDP of each respective country. Consequently, the evaluation of benefits in each county will vary in each country. Additionally, the incidence of T21 varies across different populations. As a result, it is possible that the findings of this study may potentially present an overestimation or underestimation in comparison to other studies

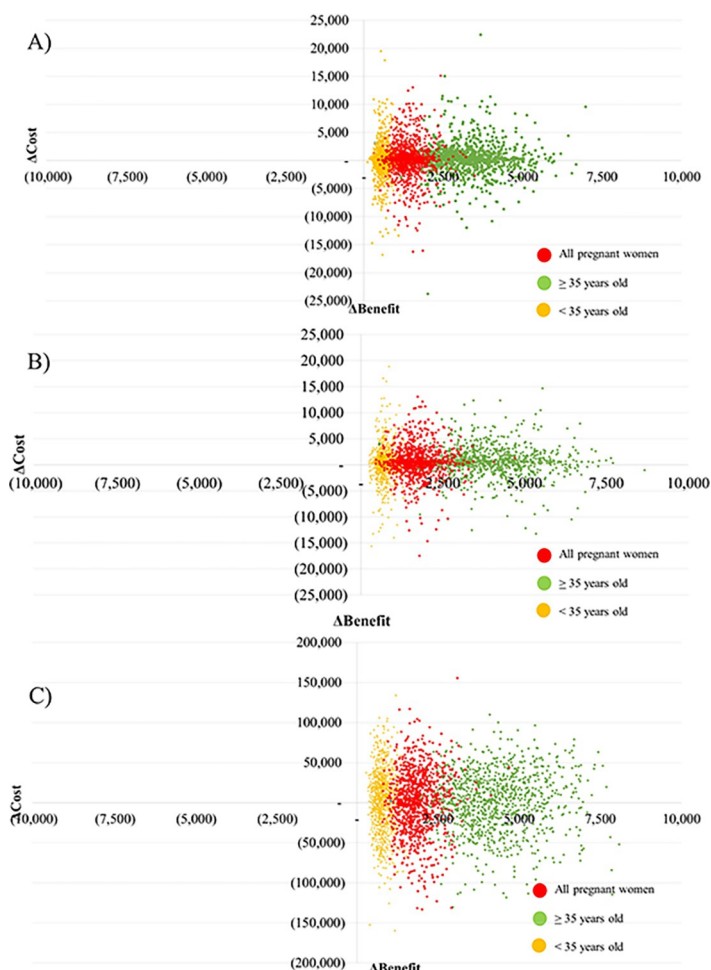

**Fig 7. Cost-benefit planes from probabilistic sensitivity analysis.** A) First-trimester screening testing, b) non-invasive prenatal testing, and c) definitive diagnosis.

conducted on the same subject matter. When compared to alternative strategies, the budget impact analysis of implementing universal NIPT as the primary screening method for all pregnant women, assuming a 100% uptake rate at the current price, yielded a total cost of 169 million USD. Based on the results obtained from the threshold sensitivity analysis, it is recommended to consider a reduction in the cost of NIPT from 247 USD to 47 USD. This

**Table 3. Threshold sensitivity analysis results based on a societal perspective.**

| Cost reduction of NIPT (%) | NIPT cost (USD) | Benefit-to-cost ratio | | |
|---|---|---|---|---|
| | | Pregnant women aged < 35 years | Pregnant women aged ≥ 35 years | Pregnant women with all ages |
| 0% | 237 | 1.30 | 6.59 | 2.88 |
| 20% | 190 | 1.41 | 7.10 | 3.11 |
| 40% | 142 | 1.54 | 7.70 | 3.37 |
| 60% | 95 | 1.68 | 8.41 | 3.68 |
| 80% | 47 | 1.86 | 9.27 | 4.06 |

NIPT, non-invasive prenatal test; USD, United States dollar

**Table 4. Budget impact analysis of each screening strategy with a 100% uptake rate.**

| Prenatal screening test | Budget impact (million USD) | | |
|---|---|---|---|
| | Pregnant women aged < 35 years | Pregnant women aged ≥ 35 years | Pregnant women with all ages |
| First-trimester test | 24 | 10 | 35 |
| Non-invasive prenatal test | 118 | 51 | 169 |
| Definitive diagnosis | 78 | 14 | 92 |

proposed adjustment would signify a significant decrease of 80%. This information can be utilized by the NHSO to engage in negotiations with laboratories offering NIPT services in order to establish a more favorable cost for the procedure.

According to several studies, it has been determined that NIPT was considered cost-effective when used as a secondary screening method. However, from a payer perspective, its high cost renders it less viable as a primary screening method. As a result, many countries have recommended NIPT to be employed as a secondary screening option [25, 26, 48–50]. However, it has been observed that NIPT is offered as a primary screening option for pregnant women, full reimbursement by the government in both the Netherlands and Australia [51, 52]. Moreover, it is worth noting that NIPT as a secondary screening is publicly funded in the United Kingdom (UK), Canada [53], and the US [49]. It is noteworthy that in Asia, NIPT is either self-financed or covered by private health insurance in China [50], Japan [54], and Taiwan [55], whereas in Hong Kong, it is offered as a second-tier screening option within the publicly funded healthcare system [50]. Due to limited resources, public funding for NIPT is not feasible in low and middle-income countries [56]. These circumstances lead to inequitable access for individuals who are unable to afford the costs. Currently, numerous laboratories in different countries have the capability to conduct NIPT within their own borders. This development has resulted in a notable reduction in the cost of NIPT, bringing it down to approximately 130–150 USD [48].

Recently, the Thai UHC has implemented a policy that provides full reimbursement for FTS or quadruple tests for all pregnant women [25]. However, it is worth noting that these tests have been associated with a high rate of false positives, leading to a subsequent need for further diagnostic procedures. In addition, this requires the establishment of additional chromosome laboratories to conduct karyotyping analysis, as well as the need for highly skilled obstetricians to carry out definitive diagnostics. However, it is important to note that these requirements entail a larger budget and an elevated risk of procedure-related loss for normal infants [26, 42]. It has been highlighted that our results indicate the potential inclusion of a universal NIPT in the prenatal screening for fetal aneuploidies within the benefit package of the UHC, if the cost of NIPT is reduced by 80% or reaches a threshold of 47 USD.

Additionally, it is imperative to acknowledge the potential implications of implementing NIPT as a widespread screening tool on the allocation of prenatal care resources. The current capacity of laboratories in Thailand to accommodate higher testing volumes and provide genetic counselling services may be limited. Presently, there are only ten public and private laboratory settings in the country that offer NIPT screening tests and genetic counselling services. Hence, it is very important to enhance the provision of genetic counselling and comprehensive information regarding NIPT to expectant mothers. This is due to the potential challenges associated with NIPT, including issues such as resampling, failure, and limited screening capabilities for certain chromosomal abnormalities [14]. Besides, it is essential that healthcare professionals receive proper training in genetic counselling to effectively communicate to pregnant women that NIPT is solely a screening test. It is also vital for these professionals to emphasize that a high-risk result from NIPT may warrant further diagnostic testing for a

definitive diagnosis. Therefore, it is imperative to furnish pregnant women with comprehensive information regarding the advantages and limitations associated with this test, enabling them to make informed decisions regarding their course of action subsequent to opting for NIPT as a screening method.

It is necessary to address the limitations of this study. Initially, the intangible benefit of pregnant women's willingness to pay for screening was not taken into consideration. This could be a room for future studies. Secondly, the current study revealed that the percentage of pregnant women over 35 years was 30%, which was higher than the prevalence of advanced maternal age in Thailand in 2018, which stood at 17% [57]. Therefore, it is possible that the occurrence rates of T13, T18, and T21 in our study might be elevated, potentially resulting in an overestimation of the budget impact. Thirdly, the data used in this study were mostly obtained from Siriraj Hospital, which is the largest university hospital in Thailand and offers NIPT services in the country. However, it may not be a representative of Thai hospitals. It is recommended that future studies should incorporate data collection from additional settings. Lastly, we refrained from investigating NIPT as a secondary screening method due to the lack of consensus regarding the appropriate cut-off levels for FTS or quadruple testing [58]. Besides, it is important to note that pregnant women are required to undergo both serum screening and NIPT as secondary screening. However, it is worth mentioning that the process of obtaining a definitive diagnosis for those who receive a high-risk result may be delayed.

## Conclusions

In summary, our study suggests that the universal NIPT as a primary screening should be implemented for all Thai pregnant women due to high detection and low positive rates compared to FTS or quadruple tests. Furthermore, it is advisable to engage in negotiations to reduce the cost of NIPT to 47 USD, in order to maximize the cost-effectiveness of this screening test. It is imperative that healthcare providers should receive comprehensive training in order to effectively educate pregnant women about NIPT. The findings would provide valuable insights for physicians in the management of chromosomal abnormalities. Additionally, they could serve as evidence-based guidance for policymakers and stakeholders involved in the development of screening policies and UHC's benefit packages within the country. Further research should be conducted in order to explore the inclusion of intangible benefits in the assessment of willingness to pay in future studies.

## Acknowledgments

The authors gratefully acknowledge Assoc. Prof. Panutsaya Tientadakul, Head of the Department of Clinical Pathology, Faculty of Medicine Siriraj Hospital, Mahidol University for her support.

## Author Contributions

**Conceptualization:** Preechaya Wongkrajang, Jiraphun Jittikoon, Wanvisa Udomsinprasert, Pattarawalai Talungchit, Sermsiri Sangroongruangsri, Saowalak Turongkaravee, Usa Chaikledkaew.

**Data curation:** Preechaya Wongkrajang.

**Formal analysis:** Preechaya Wongkrajang, Pattarawalai Talungchit, Usa Chaikledkaew.

**Investigation:** Preechaya Wongkrajang, Usa Chaikledkaew.

**Methodology:** Preechaya Wongkrajang, Jiraphun Jittikoon, Wanvisa Udomsinprasert, Pattarawalai Talungchit, Sermsiri Sangroongruangsri, Saowalak Turongkaravee, Usa Chaikledkaew.

**Project administration:** Usa Chaikledkaew.

**Resources:** Preechaya Wongkrajang.

**Supervision:** Usa Chaikledkaew.

**Validation:** Preechaya Wongkrajang, Usa Chaikledkaew.

**Visualization:** Preechaya Wongkrajang, Usa Chaikledkaew.

**Writing – original draft:** Preechaya Wongkrajang, Jiraphun Jittikoon, Wanvisa Udomsinprasert, Pattarawalai Talungchit, Sermsiri Sangroongruangsri, Saowalak Turongkaravee, Usa Chaikledkaew.

**Writing – review & editing:** Preechaya Wongkrajang, Jiraphun Jittikoon, Wanvisa Udomsinprasert, Pattarawalai Talungchit, Sermsiri Sangroongruangsri, Saowalak Turongkaravee, Usa Chaikledkaew.

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
