## [Decision Letter · Decision Letter 0]

5 Jul 2023

PONE-D-23-08966

Economic Evaluation of Prenatal Screening for Fetal Aneuploidies in Thailand

PLOS ONE

Dear Dr. Chaikledkaew,

Thank you for submitting your manuscript to PLOS ONE. After careful consideration, we feel that it has merit but does not fully meet PLOS ONE’s publication criteria as it currently stands. Therefore, we invite you to submit a revised version of the manuscript that addresses the points raised during the review process.

ACADEMIC EDITOR:

There are grammatical errors. Please get professional grammar support and provide certificates or check the article with a native english speaker

“Patau syndrome, or trisomy 13 (T13), is the third most common trisomy, affecting one in every 12,000 to 16,000 48 live births.” What is the source reference from which this sentence is taken? Please add.

“Trisomy 21 (T21), often known as Down syndrome, is the most common of these abnormalities, affecting one in every 700 to 800 live births. “ What is the source reference from which this sentence is taken? Please add.

“are currently essential for assessing the risk of a fetus [14].” correct as “are currently essential for assessing the risk of an unhealty fetus [14].”

A similar article (similar data) of a similar figure and strategy is seen in the literature (Cost Benefit Analysis of Prenatal Screening Test with Thai NIPT (Thai NonInvasive Prenatal Test) for Down Syndrome in Developing Countries Oraluck P1, Boonsong O2, Wasun C3,4, Ammarin T1 and Panyu P5*   https://www.walshmedicalmedia.com/open-access/cost-benefit-analysis-of-prenatal-screening-test-with-thai-nipt-thai-noninvasive-prenatal-test-for-down-syndrome-in-developing-cou-2375-4273-1000207.pdf). What are your views on this subject? You mentioned that it was the first, is your data from the same place?

We look forward to receiving your revised manuscript.

Kind regards,

Burak Bayraktar

Academic Editor

PLOS ONE

Journal Requirements:

Additional Editor Comments:

1) There are grammatical errors. Please get professional grammar support and provide certificates or check the article with a native english speaker

2) “Patau syndrome, or trisomy 13 (T13), is the third most common trisomy, affecting one in every 12,000 to 16,000 48 live births.” What is the source reference from which this sentence is taken? Please add.

3) “Trisomy 21 (T21), often known as Down syndrome, is the most common of these abnormalities, affecting one in every 700 to 800 live births. “ What is the source reference from which this sentence is taken? Please add.

4) “are currently essential for assessing the risk of a fetus [14].” correct as “are currently essential for assessing the risk of an unhealty fetus [14].”

5) A similar article (similar data) of a similar figure and strategy is seen in the literature (Cost Benefit Analysis of Prenatal Screening Test with Thai NIPT (Thai NonInvasive Prenatal Test) for Down Syndrome in Developing Countries Oraluck P1, Boonsong O2, Wasun C3,4, Ammarin T1 and Panyu P5* https://www.walshmedicalmedia.com/open-access/cost-benefit-analysis-of-prenatal-screening-test-with-thai-nipt-thai-noninvasive-prenatal-test-for-down-syndrome-in-developing-cou-2375-4273-1000207.pdf). What are your views on this subject? You mentioned that it was the first, is your data from the same place?

Reviewers' comments:

Reviewer's Responses to Questions

**Comments to the Author**

1. Is the manuscript technically sound, and do the data support the conclusions?

Reviewer #1: No

Reviewer #2: Yes

Reviewer #3: Yes

Reviewer #4: Yes

2. Has the statistical analysis been performed appropriately and rigorously? 

Reviewer #1: Yes

Reviewer #2: Yes

Reviewer #3: Yes

Reviewer #4: Yes

3. Have the authors made all data underlying the findings in their manuscript fully available?

Reviewer #1: Yes

Reviewer #2: Yes

Reviewer #3: Yes

Reviewer #4: Yes

4. Is the manuscript presented in an intelligible fashion and written in standard English?

Reviewer #1: Yes

Reviewer #2: Yes

Reviewer #3: Yes

Reviewer #4: Yes

5. Review Comments to the Author

Reviewer #1: Thank you for your efforts on this project. This is a very fascinating paper.

I would recommend accepting this project for publication.

Just a minor correction: Please refer to (Miscarriage) if you mean a spontaneous early pregnancy loss, rather than abortion if you imply an early pregnancy termination.

Reviewer #2: This is a well conducted study emphasizing the need for a universal screening program to be implemented for prenatal screening. NIPT seems to be the superior strategy due to its high detection rate and low false positive rates. Its cost is also expected to be further reduced if it will be used more broadly. These data need to be taken into consideration by national policymakers and stakeholders in the development of screening policies as well as health benefits packages. I support the publication of this article.

Reviewer #3: Thank you for submitting your manuscript entitled "Economic Evaluation of Prenatal Screening for Fetal Aneuploidies in Thailand." I have carefully reviewed the manuscript and would like to provide you with constructive feedback to strengthen your work. Overall, the study presents valuable insights into the cost-benefit analysis of prenatal screening tests and their implications for managing chromosomal abnormalities. Below are my specific comments:

Introduction:

The introduction effectively highlights the significance of chromosomal abnormalities, especially trisomy 21, and the associated healthcare burden. However, it would be helpful to provide a brief overview of the current prenatal screening practices and their limitations, to better contextualize the need for cost-benefit analysis.

Methods:

The description of the target population and data sources is clear and informative. However, it would be beneficial to provide more details on the decision-analytic model used for the cost-benefit analysis, including the specific variables and assumptions considered.

Additionally, please clarify how the three groups (all ages, older than 35 years, and less than 35 years) were selected for evaluating each screening strategy.

Results:

The results section provides comprehensive information on the cost-benefit outcomes of the screening strategies. However, it would be useful to include a discussion on the specific factors that contribute to the higher net benefits and cost-effectiveness of certain strategies, such as the reasons behind the lower cost of FTS and the higher cost of definitive diagnosis.

Discussion:

The discussion provides a thorough analysis of the cost-benefit results and their implications. It effectively addresses the practical limitations of FTS and emphasizes the advantages of NIPT in terms of accuracy and reduced false-positive rates.

However, it would be valuable to discuss the potential impact of implementing NIPT as a universal screening tool on prenatal care resources, including the capacity of laboratories to handle increased testing volumes and the availability of genetic counseling services.

Conclusion:

The conclusion effectively summarizes the key findings and emphasizes the importance of implementing NIPT as a universal screening tool. It provides a clear call to action for physicians, policymakers, and stakeholders.

To further strengthen the conclusion, it would be beneficial to briefly mention any recommendations or future directions for research, policy development, or healthcare system adaptations based on the study's findings.

Overall, this manuscript makes a valuable contribution to the field of prenatal screening for chromosomal abnormalities. I believe addressing the minor points mentioned above will enhance the clarity and impact of your work. Once again, thank you for submitting your manuscript to our journal, and I look forward to seeing the revised version.

Reviewer #4: 1. ABSTRACT: The abstract should be structured into Background, Objective, Methods, Results and Conclusion.

There appear to be too many key words for the abstract. It will be good to reduce the key words to the most important words that will be used for a literature search.

2. Why were women less than 35 years of age screened for aneuploidies? Is it part of the hospital guidelines or protocol?

3. It may be good to put a foot note explaining what beta, gamma and log normal means in table 1.

4. You may need to further defend your conclusion on making NIPT a universal screening tool since other methods of screening equally have their advantages.

6. PLOS authors have the option to publish the peer review history of their article (what does this mean?). If published, this will include your full peer review and any attached files.

Reviewer #1: **Yes: **Mena Abdalla

Reviewer #2: **Yes: **Nikolaos Antonakopoulos

Reviewer #3: **Yes: **Shinnosuke Komiya

Reviewer #4: No

---

## [Author Response · Author response to Decision Letter 0]

27 Aug 2023

Response to Academic Editor & Journal requirements & Reviewer:

Academic Editor

1. There are grammatical errors. Please get professional grammar support and provide certificates or check the article with a native English speaker.

Response: Thank you very much. This article has already been checked with a native English speaker. 

2.“Patau syndrome, or trisomy 13 (T13), is the third most common trisomy, affecting one in every 12,000 to 16,000 live births.” What is the source reference from which this sentence is taken? Please add.

Response: Thank you very much. We have added a reference as suggested. 

(Introduction, line 55, page 3) 

“Patau syndrome, also known as trisomy 13 (T13), is a relatively prevalent trisomy condition, occurring in approximately one out of every 12,000 live births [1].” 

3.“Trisomy 21 (T21), often known as Down syndrome, is the most common of these abnormalities, affecting one in every 700 to 800 live births. “What is the source reference from which this sentence is taken? Please add.

Response: Thank you very much. We have added a reference as suggested. 

(Introduction, line 48, page 3) 

“Trisomy 21 (T21), often known as Down syndrome, is the most common of these abnormalities, occurring one in every 700 live births [1].” 

4.“are currently essential for assessing the risk of a fetus [14].” correct as “are currently essential for assessing the risk of an unhealthy fetus [14].”

Response: Thank you very much. We have corrected as suggested.

(Introduction, line 63, page 3) 

“Prenatal screening tests, including serum screening, nuchal translucency (NT) from ultrasound, and genetic screening, are currently essential for assessing the potential risk of an unhealthy fetus [14].” 

5. A similar article (similar data) of a similar figure and strategy is seen in the literature (Cost Benefit Analysis of Prenatal Screening Test with Thai NIPT (Thai NonInvasive Prenatal Test) for Down Syndrome in Developing Countries Oraluck P1, Boonsong O2, Wasun C3,4, Ammarin T1 and Panyu P5* https://www.walshmedicalmedia.com/open-access/cost-benefit-analysis-of-prenatal-screening-test-with-thai-nipt-thai-noninvasive-prenatal-test-for-down-syndrome-in-developing-cou-2375-4273-1000207.pdf). What are your views on this subject? You mentioned that it was the first, is your data from the same place?

Response: Thank you very much for raising this point. We have added more explanation why our study is different from Oraluck et al’s study in “Discussion” part. 

(Discussion, paragraph 4, page 19-20)

“In contrast to our study, two previously published cost-benefit studies conducted in Thailand [28, 29] suggest that implementing a universal NIPT as the primary screening method for Down syndrome (T13) was not deemed cost-beneficial. However, these studies ascertained that utilizing NIPT as a secondary screening approach was the most cost-beneficial choice. It should be noted that the discrepancies in our results can be ascribed to the following reasons. First, we conducted an analysis on the cost-benefit of implementing a universal NIPT for Down syndrome (T13), Edwards syndrome (T18), and Patau syndrome (T21), while two aforementioned studies specifically focused on Down syndrome (T13) exclusively. Apart from this, the majority of the data, such as prevalence, incidence, uptake rate, and costs, were obtained from Siriraj Hospital, while a study by Oraluck et al. [28] primarily relied on the findings of Pattanaphesaj et al [31], which were conducted a decade ago. Besides this, a study by Wanapirak et al. [29] primarily utilized most of the data from their previous study [22]. Consequently, our study did not coincide with the findings of the previously mentioned studies.”

Response: Thank you very much. We have already checked that our manuscript meets PLOS ONE’s style requirements. 

Reviewer #1: 

Thank you for your efforts on this project. This is a very fascinating paper.

I would recommend accepting this project for publication.

Just a minor correction: Please refer to (Miscarriage) if you mean a spontaneous early pregnancy loss, rather than abortion if you imply an early pregnancy termination.

Response: Thank you very much for taking your time for reviewing our manuscript. We feel that the revised manuscript is much further improved as a consequence of your inputs. We have changed from “abortion” to “miscarriage” throughout our manuscript, when we meant a spontaneous early pregnancy loss or implied an early pregnancy termination.

Reviewer #2: 

This is a well conducted study emphasizing the need for a universal screening program to be implemented for prenatal screening. NIPT seems to be the superior strategy due to its high detection rate and low false positive rates. Its cost is also expected to be further reduced if it will be used more broadly. These data need to be taken into consideration by national policymakers and stakeholders in the development of screening policies as well as health benefits packages. I support the publication of this article.

Response: Thank you very much for taking your time for reviewing our manuscript. We feel that the revised manuscript is much further improved as a consequence of your inputs. 

Reviewer #3: 

Thank you for submitting your manuscript entitled "Economic Evaluation of Prenatal Screening for Fetal Aneuploidies in Thailand." I have carefully reviewed the manuscript and would like to provide you with constructive feedback to strengthen your work. Overall, the study presents valuable insights into the cost-benefit analysis of prenatal screening tests and their implications for managing chromosomal abnormalities. Below are my specific comments:

Response: Thank you very much for taking your time for reviewing our manuscript. We feel that the revised manuscript is much further improved as a consequence of your inputs. 

1. Introduction:

The introduction effectively highlights the significance of chromosomal abnormalities, especially trisomy 21, and the associated healthcare burden. However, it would be helpful to provide a brief overview of the current prenatal screening practices and their limitations, to better contextualize the need for cost-benefit analysis.

Response: Thank you very much for great suggestion. We have added a brief overview of the current prenatal screening practices and their limitations as follows.

(Introduction, lines 99-115, page 5-6)

“Since 2016, the Subcommittee on Health Promotion and Disease Prevention under the National Health Security Office (NHSO) has included Down syndrome screening in the UHC’s benefit package for pregnant women aged 35 years and over [25]. Later since 2022, the Subcommittee has implemented an expansion of the quadruple test for pregnant women across all age groups, commonly referred to quadruple for all [25]. Consequently, the implementation of Down Syndrome Prevention and Control Pilot Program in 2016 across five provinces yielded a high acceptance rate of quadruple screening, but resulted in a false positive rate of 4-10%, potentially leading to a considerable number of unnecessary amniocentesis procedures [26]. As of the time, there is a lack of available cost-effectiveness data regarding the optimal prenatal screening test for T13, T18, and T21, as well as the recommended age group of pregnant women for screening. Therefore, the NHSO has formally requested the aforementioned information to facilitate informed decision-making regarding the potential implementation of a universal prenatal screening test policy, which would encompass all pregnant women. Accordingly, the objective of this study was to conduct a cost-benefit analysis of different prenatal chromosomal abnormality screening tests for detecting T13, T18, and T21 in all pregnant women classified into three age groups: all ages, older than 35 years, and less than 35 years.”

2.Methods:

The description of the target population and data sources is clear and informative. However, it would be beneficial to provide more details on the decision-analytic model used for the cost-benefit analysis, including the specific variables and assumptions considered. 

Response: Thank you very much for very helpful suggestion. We have provided more details on the decision-analytic model used as follows.

(Model structure, line 128-156, page 6-7)

“A decision-analytic model was developed to conduct a cost-benefit analysis of three screening strategies: 1) FTS, 2) NIPT, and 3) definitive diagnosis, in comparison to no screening. Regarding FTS option (Figure 1), pregnant women are provided with the opportunity to make an informed decision regarding their acceptance or declination of the FTS test. If the test is accepted, it is possible for the results to detect the presence of T13, T18, and T21 in the fetuses, either through true positive or false negative results. If the individuals received true positive test results and accepted definitive diagnosis, they might have procedure related abortion. For those with true positive results, they might choose to undergo pregnancy termination, deliver live births with chromosomal abnormalities such as T13, T18, and T21, or experience spontaneous abortion. For those with false negative results or those denying definitive diagnosis, they might deliver live births with T13, T18, and T21 or have spontaneous abortion. On the other hand, for those having non-trisomy detected by false positive result, they would choose to either accept or reject the definitive diagnosis. For those with true negative results, they would either give birth to infants without trisomy or experience spontaneous abortion. However, in the event that they decline the FTS test, there is a possibility of delivering live births without trisomy or experiencing spontaneous abortion.

In the context of a universal NIPT alternative (Figure 2), pregnant women would have the option to either accept or deny the utilization of NIPT. If the NIPT results were positive, it would be possible that the fetuses might have been identified with T13, T18, and T21, or non-trisomy conditions, as determined by either successful or unsuccessful NIPT outcomes. For those with NIPT failure, they would be provided with a conclusive diagnosis. In cases where NIPT yields successful results, it is possible to detect fetuses with T13, T18, and T21 through either false positive or true negative finding. Conversely, in instances where NIPT is unsuccessful, it is possible to detect non-trisomy fetuses through either false positive or true negative finding. The decision tree model pathways for definitive diagnosis, true positive, false negative, false positive, and true negative in NIPT were found to be comparable to those observed in FTS. Figure 3A and 3B demonstrates definitive diagnosis or amniocentesis for T13, T18, and T21 as well as non-trisomy, respectively. The data analysis was performed using Microsoft Excel 2019 (Microsoft, WA, USA).”

Response: Thank you very much for very helpful suggestion. We have provided more details on the specific variables and assumptions considered as follows.

(Indirect cost, line 196-206, page 9)

“The productivity loss or income loss was estimated by working age range multiplied by the Thai Gross Domestic Product (GDP) per capita per year (6,370.8 USD) [35]. It was assumed that the working age range was 45 years (60-15 years). In addition, based on the recommendation from the Thai HTA guidelines, since cost values are different in different time periods, future values of total expected productivity loss (FV) should be adjusted to present values (PV) using an annual discount rate of 3% [36] based on this formula: PV = FV x [1/(1+r)n], where PV = present value, FV = future value, r = discount rate, and n = each year in the future [36]. Moreover, we also assumed that expected income was increased by 4% per year, which was obtained from an annual income growth rate during 1990-2022 in Thailand [37].”

(Direct benefit, line 212-215, page 10)

“It was assumed that the avoided healthcare costs were equal to the average total healthcare costs of patients with T13, T18, and T21 which were retrieved from previous studies by Pattanaphesaj et al [31] and Walker et al [38].”

(Indirect benefit, line 217-224, page 10)

“An indirect benefit was the productivity gain of caregivers who did not have to take care of trisomy children as a result of prenatal screening test. Human capital approach was applied by multiplying the average expected survival of trisomy patients with the caregivers’ expected income referred from the annual Thai GDP per capita per year (6,370.8 USD) [35]. We assumed that the survival of fetuses with T13 and T18 was one year [39], while that of those with T21 was 50 years [40]. For caregivers of trisomy children with T21, future values of total expected productivity gain were adjusted to their present values using the discount rate of 3% [36] and income growth rate of 4% per year [37].”

Additionally, please clarify how the three groups (all ages, older than 35 years, and less than 35 years) were selected for evaluating each screening strategy.

Response: Thank you very much for very useful suggestion. We have clarified in “Material and methods” section as follows.

(Target population, line 122-124, page 6)

“These three groups with a cut-off age at 35 years were selected for evaluating each screening strategy based on the recommendation of the American College of Obstetricians and Gynecologists Clinical Practice Guidelines [27].”

3.Results:

The results section provides comprehensive information on the cost-benefit outcomes of the screening strategies. However, it would be useful to include a discussion on the specific factors that contribute to the higher net benefits and cost-effectiveness of certain 

strategies, such as the reasons behind the lower cost of FTS and the higher cost of definitive diagnosis.

Response: Thank you very much for great suggestions. We have added more details as suggested. 

(Discussion, paragraph 1, line 317-323, page 17-18)

“The findings of this study revealed that all three methods were cost-beneficial based on the governmental perspective. Among these methods, FTS was identified as the most cost-beneficial option, particularly for pregnant women aged over 35 years. The cost of FTS was significantly lower (27.5 USD) compared to NIPT (237 USD). However, when considering the societal perspective, it is generally not considered cost-beneficial to pursue a definitive diagnosis, as this often led to additional indirect costs due to the loss of a pregnancy in typical cases.”

4.Discussion:

The discussion provides a thorough analysis of the cost-benefit results and their implications. It effectively addresses the practical limitations of FTS and emphasizes the advantages of NIPT in terms of accuracy and reduced false-positive rates.

However, it would be valuable to discuss the potential impact of implementing NIPT as a universal screening tool on prenatal care resources, including the capacity of laboratories to handle increased testing volumes and the availability of genetic counseling services.

Response: Thank you very much for your very useful suggestion. We have added more explanation as suggested in “Discussion” part.

(Discussion, paragraph 8, line 421-436, page 22)

“Additionally, it is imperative to acknowledge the potential implications of implementing Non-Invasive Prenatal Testing (NIPT) as a widespread screening tool on the allocation of prenatal care resources. The current capacity of laboratories in Thailand to accommodate higher testing volumes and provide genetic counselling services may be limited. Presently, there are only ten public and private laboratory settings in the country that offer NIPT screening tests and genetic counselling services. Hence, it is very important to enhance the provision of genetic counselling and comprehensive information regarding NIPT to expectant mothers. This is due to the potential challenges associated with NIPT, including issues such as resampling, failure, and limited screening capabilities for certain chromosomal abnormalities [14]. Besides, it is essential that healthcare professionals receive proper training in genetic counselling to effectively communicate to pregnant women that NIPT is solely a screening test. It is also vital for these professionals to emphasize that a high-risk result from NIPT may warrant further diagnostic testing for a definitive diagnosis. Therefore, it is imperative to furnish pregnant women with comprehensive information regarding the advantages and limitations associated with this test, enabling them to make informed decisions regarding their course of action subsequent to opting for NIPT as a screening method.”

5.Conclusion:

The conclusion effectively summarizes the key findings and emphasizes the importance of implementing NIPT as a universal screening tool. It provides a clear call to action for physicians, policymakers, and stakeholders.

To further strengthen the conclusion, it would be beneficial to briefly mention any recommendations or future directions for research, policy development, or healthcare system adaptations based on the study's findings.

Response: Thank you very much for your great suggestion. We have added the sentences in “Conclusion” as suggested.

(Conclusion, line 453-463, page 23)

“In summary, our study suggests that the universal NIPT as a primary screening should be implemented for all Thai pregnant women due to high detection and low positive rates compared to FTS or quadruple tests. Furthermore, it is advisable to engage in negotiations to reduce the cost of NIPT to 47 USD, in order to maximize the cost-effectiveness of this screening test. It is imperative that healthcare providers should receive comprehensive training in order to effectively educate pregnant women about NIPT. The findings would provide valuable insights for physicians in the management of chromosomal abnormalities. Additionally, they could serve as evidence-based guidance for policymakers and stakeholders involved in the development of screening policies and UHC’s benefit packages within the country. Further research should be conducted in order to explore the inclusion of intangible benefits in the assessment of willingness to pay in future studies.”

Reviewer #4: 

1. ABSTRACT: The abstract should be structured into Background, Objective, Methods, Results and Conclusion.

There appear to be too many key words for the abstract. It will be good to reduce the key words to the most important words that will be used for a literature search.

Response: Thank you very much for pointing this out. After checking carefully, we have added “Background” to make the abstract structured as suggested. 

(Abstract, 23-27, page 2)

“Historically, there has been a lack of cost-effectiveness data regarding the inclusion of universal non-invasive prenatal testing (NIPT) for trisomy 21, 18, and 13 in the benefit package of the Universal Health Coverage (UHC) in Thailand. Therefore, this study aimed to perform the cost-benefit analysis of prenatal screening tests and calculate the budget impact that would result from the implementation of a universal NIPT program.”

In addition, we already provided three key words i.e., trisomy, economic evaluation and cost-benefit analysis as indicated on page 1.

2. Why were women less than 35 years of age screened for aneuploidies? Is it part of the hospital guidelines or protocol?

Response: Thank you very much for raising this point. Pregnant women less than 35 years of age were screened for aneuploidies based on the recommendation of the American College of Obstetricians and Gynecologists Clinical Practice Guidelines. We have clarified in “Material and methods” section as follows.

(Target population, line 122-124, page 6)

“These three groups with a cut-off age at 35 years were selected for evaluating each screening strategy based on the recommendation of the American College of Obstetricians and Gynecologists Clinical Practice Guidelines [27].”

3. It may be good to put a foot note explaining what beta, gamma and log normal means in table 1.

Response: Thank you very much for great suggestion. We have added a foot note explaining what beta, gamma and log normal means in table 1 as suggested.

(Table 1, page 12)

“*Beta, log-normal, or gamma distribution is appropriate for parameter values ranging from 0 to 1, 0 to ∞, and > 0 to ∞, respectively.”

4. You may need to further defend your conclusion on making NIPT a universal screening tool since other methods of screening equally have their advantages.

Response: Thank you very much for great suggestion. We have added the sentences as suggested in “Conclusion”.

(Conclusion, line 453-463, page 23)

“In summary, our study suggests that the universal NIPT as a primary screening should be implemented for all Thai pregnant women due to high detection and low positive rates compared to FTS or quadruple tests. Furthermore, it is advisable to engage in negotiations to reduce the cost of NIPT to 47 USD, in order to maximize the cost-effectiveness of this screening test. It is imperative that healthcare providers should receive comprehensive training in order to effectively educate pregnant women about NIPT. The findings would provide valuable insights for physicians in the management of chromosomal abnormalities. Additionally, they could serve as evidence-based guidance for policymakers and stakeholders involved in the development of screening policies and UHC’s benefit packages within the country. Further research should be conducted in order to explore the inclusion of intangible benefits in the assessment of willingness to pay in future studies.”

---

## [Decision Letter · Decision Letter 1]

4 Sep 2023

Economic Evaluation of Prenatal Screening for Fetal Aneuploidies in Thailand

PONE-D-23-08966R1

Dear Dr. Chaikledkaew,

We’re pleased to inform you that your manuscript has been judged scientifically suitable for publication and will be formally accepted for publication once it meets all outstanding technical requirements.

Kind regards,

Burak Bayraktar

Academic Editor

PLOS ONE

Additional Editor Comments (optional):

Reviewers' comments:

Reviewer's Responses to Questions

**Comments to the Author**

1. If the authors have adequately addressed your comments raised in a previous round of review and you feel that this manuscript is now acceptable for publication, you may indicate that here to bypass the “Comments to the Author” section, enter your conflict of interest statement in the “Confidential to Editor” section, and submit your "Accept" recommendation.

Reviewer #1: All comments have been addressed

Reviewer #3: All comments have been addressed

2. Is the manuscript technically sound, and do the data support the conclusions?

Reviewer #1: Yes

Reviewer #3: Yes

3. Has the statistical analysis been performed appropriately and rigorously? 

Reviewer #1: Yes

Reviewer #3: Yes

4. Have the authors made all data underlying the findings in their manuscript fully available?

Reviewer #1: Yes

Reviewer #3: Yes

5. Is the manuscript presented in an intelligible fashion and written in standard English?

Reviewer #1: Yes

Reviewer #3: Yes

6. Review Comments to the Author

Reviewer #1: Thank you very much for the authors for addressing all the previous reviewers comments and made the required modifications. With best wishes

Reviewer #3: Your paper offers a comprehensive analysis of the cost-benefit aspects of various prenatal screening methods for chromosomal abnormalities, particularly in the context of Thailand. The depth of the analysis, especially in examining societal and governmental perspectives, is commendable.

7. PLOS authors have the option to publish the peer review history of their article (what does this mean?). If published, this will include your full peer review and any attached files.

Reviewer #1: **Yes: **Mena Abdalla

Reviewer #3: No

---

## [Editor Report · Acceptance letter]

8 Sep 2023

PONE-D-23-08966R1 

Economic evaluation of prenatal screening for fetal aneuploidies in Thailand 

Dear Dr. Chaikledkaew:

I'm pleased to inform you that your manuscript has been deemed suitable for publication in PLOS ONE. Congratulations! Your manuscript is now with our production department. 

Kind regards, 

on behalf of

Dr. Burak Bayraktar 

Academic Editor

PLOS ONE